# Frontal Regions and Executive Function Testing: A Doubted Association Shown by Brain-Injured Patients

**DOI:** 10.3390/neurosci6040105

**Published:** 2025-10-14

**Authors:** Demis Basso, Ida Bosio, Vincenza Tarantino, Francesco Carabba

**Affiliations:** 1Cognitive and Educational Sciences Lab (CESLab), Faculty of Education, Free University of Bolzano-Bozen, 39042 Bressanone-Brixen, BZ, Italy; francesco.carabba@unibz.it; 2Department of General Psychology, University of Padua, 35131 Padova, Italy; ida.bosio@studenti.unipd.it; 3Department of Psychology, Educational Sciences and Human Movement, University of Palermo, 90128 Palermo, Italy; vincenza.tarantino03@unipa.it

**Keywords:** executive functions, cortical areas, frontal lobe, neuropsychology

## Abstract

Since its introduction, the construct of executive functions (EFs) has been associated with a set of tests to assess these functions and a brain network centered in the associative frontal brain regions. While the majority of perspectives have endorsed these associations, some studies have started casting doubts on them. In this article, the association between the construct of EFs, the tests used to assess them, and the involvement of frontal regions is examined. A sample of 28 patients with brain injuries was divided into three subgroups according to the region of the injury (anterior, posterior, antero-posterior). Patients were assessed with a battery of tests, including 25 measures of EFs and 6 control measures. A series of regression models revealed no significant differences in performance across the three groups. Findings indicate that the EF tests are not specific enough to differentiate EFs and brain injuries. The alleged reference of EFs to the frontal areas of the brain should attribute a higher role to other associative areas. The present study provides recommendations about how the EFs concept could be improved through methodological refinements and/or its dissemination.

## 1. Introduction

Among the most prominent concepts developed in neuroscience, executive functions (EFs) are central in several subfields like cognitive and clinical research. While a universal consensus has not been reached, the term usually designates a few higher-order cognitive processes that coordinate simpler cognitive abilities such as orienting attention, motor preparation, and sensory processing. EFs generate considerable interest due to their relevance to brain injury and their critical role in goal-directed behavior, adaptation to novel situations, and behavioral regulation. However, despite the substantial advances made over recent decades by cognitive and neuropsychological research, EFs belong to the list of the “gray areas” that remain in our understanding of human cognition. The construct of EFs has evolved substantially since its early formulation, and many authors have proposed varying definitions. They still present considerable challenges, particularly regarding their conceptual definition, reliable assessment, and precise neuroanatomical localization.

Initially, EFs were often conceptualized as a single unitary process, similarly to the general intelligence factor “g” [1,2,3,4] or working memory [5]. However, subsequent research has demonstrated that executive functioning is best understood as a set of interrelated but separable processes [6,7,8]. Empirical efforts sought to clarify the structure of EFs through statistical modeling approaches. Using Confirmatory Factor Analysis (CFA) and Structural Equation Modeling, Miyake and collaborators [9] examined performance across a battery of tasks designed for frequently postulated EFs. Their analysis focused on three constructs: mental set shifting (shifting), information updating and monitoring in working memory (updating), and inhibition of prepotent responses (inhibition). The CFA results indicated that these three components are moderately correlated but separable, each contributing uniquely to task performance while also sharing a degree of common variance. Building on this hypothesis of unity and diversity, a review by [10] highlighted how the pattern of EFs, sharing an underlying commonality but also being separable, has been replicated numerous times across different ages and populations. The review also incorporated evidence from genetic studies suggesting heritable influences on both common and specific variances of EFs, eventually supported by neuroimaging findings. More recently, ref. [11] conducted a systematic review and re-analyzed latent variable studies on executive functions across the lifespan. They confirmed that inhibition, updating/working memory, and shifting are the most frequently evaluated constructs, largely following the model proposed by [9]. They also found evidence for increasing diversity of executive functions over the course of development, particularly from childhood into adulthood, and potentially slight de-differentiation in older age.

Overall, almost all theorizations share some core components such as inhibition, planning, and control, but many also include other processes. This variety in conceptualization leads to suspicion about the nature of the concept itself. According to some scholars [9,10,11,12], EFs are still considered to be an elusive concept, though the challenges might be overcome by collecting further data and elaborating more precise models of functioning. To face this criticism, it is suggested that theorization should include the methods used to collect data. Other perspectives criticize the concept of EFs by arguing against the strict localization in the frontal areas of the brain. The complexity of studying EFs can be addressed by focusing on (a) methodological limitations inherent in commonly used tasks, and (b) the neuroanatomical debate concerning the extent to which EF performance can be localized to discrete brain regions.

### 1.1. Limits of Executive Function Tests

Despite their central role in neuropsychological assessment protocols, the tasks identified as aimed at studying EFs present a few challenges. The most common paradigms used to assess executive functions are listed in Table 1 with the components generally associated with each task, shown in the leftmost column, and the seminal paper that presented the task. The list aims to offer a representative view of the tasks used for each process included in the EFs definition. It should be noted that the great majority of the most important tasks were introduced more than 30 years ago, while the majority of theories on EFs were developed later. Therefore, a mismatch exists between the constructs tested and the constructs postulated [13].

Many tasks are not considered process-pure and often involve multiple overlapping cognitive functions [8,9,46]. This overlap stems from both the complexity of the behavior involved and the multifaceted nature of most tasks used to assess EFs [47]. This has raised concerns about the extent to which such tasks can reliably distinguish EFs with respect to distinct executive sub-processes or isolate executive functioning from other domains (see [6,11]). For instance, although the WCST [30,48] is often used to assess cognitive flexibility and set shifting, its successful performance also requires working memory, error monitoring, and hypothesis testing [49,50,51]. Neuroimaging and lesion studies further revealed that performance on the WCST is supported by a distributed network involving both frontal and non-frontal regions, including the basal ganglia and parietal cortex [50,52]. As a result, the WCST is best described as a sensitive, but not specific, indicator of “frontal lobe functioning” [53]. A similar issue arises with the Stroop Color-Word Test [36], long considered a prototypical measure of inhibitory control. Although it reliably elicits interference effects and is widely used to infer frontal dysfunction, evidence linking Stroop performance to focal frontal damage is surprisingly sparse [53]. Studies have shown that parietal areas may also contribute, perhaps via mechanisms related to selective attention or sensorimotor integration [54,55]. Thus, even well-established EFs tasks fail to isolate the EFs they purport to measure. These concerns extend to the domain of planning. Traditional planning tasks, such as the Tower of London [45] and the Elithorn’s Perceptual Maze [43], have been widely adopted in clinical and experimental settings. Yet, research has questioned whether these tasks truly capture planning as a distinct executive process. For example, while the Tower of London requires preplanning and working memory, studies suggest that its reliance on structured, abstract problems may not generalize well to real-world planning demands [56,57]. Similarly, the Elithorn Maze involves aspects of visual search and rule maintenance, blurring its functional specificity [58]. This has led researchers to explore alternative approaches that emphasize ecological validity and the dynamic nature of planning. Tasks based on the Traveling-Salesperson Problem [59], such as the City Map Test [60], Plan-A-Day [61], and the Maps test [62], aim to model everyday problem-solving by requiring participants to generate strategies under open-ended constraints. These tasks demonstrate that successful planning involves not just mental simulation and goal management, but also the ability to flexibly update actions during execution, which is an aspect often missing from traditional paradigms. Based on these studies, the issue is confirmed: none of these tasks cleanly isolates either planning or its executive components. Instead, they underscore the distributed and integrative nature of executive control. This complexity may partly explain the ongoing debate regarding the structure of EFs and their neural bases.

In recent years, the development of immersive and interactive technologies such as virtual reality (VR) has also opened new avenues for EFs assessment and offers new insights into cognitive processes. VR-based tools can simulate complex, dynamic environments that more closely approximate real-world cognitive demands. These new instruments hold promises for enhancing ecological validity while also allowing fine-grained control over task parameters and performance monitoring. Tools such as the Virtual Multiple Errands Test replicate complex real-life tasks (e.g., shopping with constraints and interruptions) within virtual settings, capturing aspects of planning, task-switching, inhibition, and strategy use with greater ecological fidelity than traditional assessments [63]. Unlike static paper-and-pencil tasks, VR systems allow researchers to track user actions, rule violations, and strategies in real time, offering detailed behavioral profiles that are often more sensitive to subtle executive dysfunctions, particularly in populations where conventional tests fail to detect impairments [63]. As shown in the systematic review by Borgnis and colleagues [64], VR-based tools have demonstrated promising psychometric properties (e.g., construct validity, usability, and ecological validity) and have already been successfully applied in both neurological and psychiatric populations. Recent studies utilizing VR have further challenged the traditional view of EFs as being strictly localized in designated frontal associative brain areas and suggest that EFs may engage broader neural networks [65].

In summary, although a great number of tasks and emerging approaches are available to study EFs, their interpretative power is constrained by the multidimensionality of task demands and the distributed nature of neural networks supporting executive control. While each process may be assessed by many tasks, each task cannot provide information on only one process at a time, and this ambiguous correspondence between test performance and process specificity continues to fuel debates on the structure and ontological status of EFs.

### 1.2. Anatomical Localization of Executive Functions

For decades, it has been assumed that the control of cognitive processes is localized in the frontal associative regions of the brain. This perspective emerged largely from early neuropsychological observations: patients with frontal lobe damage often displayed marked impairments in these domains and failures in a variety of neuropsychological measures [66,67]. As a result, executive deficits were long assumed to reflect dysfunction of the frontal lobes. However, this localizationist view has been challenged because of the complexity and heterogeneity of the PFC itself.

Different subregions seem to contribute uniquely to executive processes, and anatomical studies support functional differentiation within the PFC. The DLPFC is generally linked to working memory and planning, whereas the vmPFC and orbitofrontal cortex (OFC) are implicated in emotional processing, social decision-making, and value-based reasoning [68]. These functional distinctions, however, are not cleanly mapped onto specific behavioral tasks, making it difficult to draw one-to-one correspondences between lesion site and cognitive deficit. To address these challenges, contemporary theories usually favor a network-based model of executive function, in which EFs arise from distinct but interconnected large-scale brain networks that extend beyond the frontal lobes [69]. Neuroimaging and lesion studies have identified six major networks implicated in EF and modulated based on task demands [70]: the frontoparietal control network, the salience network, the cingulo-opercular network, the dorsal and ventral attention networks, and the default mode network. Moreover, EFs have been increasingly described through the lens of hot and cold cognition, which distinguishes between emotionally neutral (cold) and emotionally laden (hot) executive processes. Recent scientific literature proposes that cold EFs (e.g., working memory, response inhibition) are primarily associated with lateral PFC and dorsal anterior cingulate cortex (dACC), while hot EFs (e.g., decision-making under uncertainty, emotional regulation) are more dependent on medial/orbitofrontal PFC and ventral ACC, as well as the posterior cingulate cortex [71]. While these domains are not strictly separable, this model emphasizes that the emotional context of a task modulates the specific networks involved. The adaptive coding hypothesis [72] offers a complementary perspective, proposing that neurons in the PFC are not hardwired for specific tasks but can flexibly adapt their properties depending on task demands. In this view, executive processes such as working memory, attention, and control are seen as different manifestations of a common underlying neural mechanism, in which the PFC is the key resource that provides “such [cognitive] operations greater focus, power or flexibility” [72] (p. 827).

Considering these developments, the idea of a centralized executive agent seated within the frontal lobes has been largely abandoned. As [73] argued, the PFC should no longer be conceptualized as the sole source of executive control, but as part of a flexible, context-sensitive system integrating diverse cognitive domains. Executive dysfunction, therefore, cannot be attributed to a single lesion or mechanism, but must be understood as the disruption of coordinated processes across a distributed and dynamic brain network. In turn, this aspect could also explain why many of the so-called “frontal lobe tests” have failed to reliably differentiate patients labelled with dysexecutive symptoms, showing an associated frontal damage, from those showing lesions elsewhere in the brain [47,74]. Several studies have demonstrated that performance deficits in classical executive tasks, such as the Wisconsin Card Sorting test or verbal fluency tasks, are not uniquely associated with frontal lesions [75,76], providing support to the view of an executive control distributed on neural networks rather than found on a single cortical region.

### 1.3. The Present Study

Recent findings suggest that there is neither a single frontal lobe syndrome nor a one-to-one functional and anatomical correspondence. However, the debate about the localization of EFs remains open, given that none of the competing views has clearly prevailed. The present study was conceived to evaluate the extent of the association between EFs, PFC, and “frontal lobe tests”. Patterns of performance on EF and non-EF tasks were compared across patients with lesions localized to anterior, posterior, or both cortical regions. The results will provide evidence whether EF measures reflect specific frontal lobe dysfunction or indicate broader impairment in the associative network. It was hypothesized that the concept of EFs would confirm the mainstream literature if at least some of the tests used to assess EFs were able to differentiate the performance of the three groups of patients. In particular, an effect is expected on the three EFs identified by the modal model [9]: working memory, shifting, and inhibition. If this pattern did not emerge, then the commonly assumed triadic relationship linking brain lesion, cognitive function, and EF test performance is likely to be too simplistic to account for the complexity of real-world clinical data [53,77,78]. While the EF models based on clear brain localization may offer a useful heuristic for initial clinical reasoning, they often fall short when tested against actual performance from patients with brain lesions. Many studies currently use correlation techniques (like electro-encephalography and functional Magnetic Resonance Imaging), but, despite their richness in time and/or space precision, these are not able to clarify causal relationships. Although the classical neuropsychological approach cannot provide conclusive results because of the lack of experimental control, it is still useful to (in-)validate models. For this reasons, the present study involves brain-injured patients studied through a wide set of tests.

## 2. Materials and Methods

### 2.1. Participants

Thirty-nine traumatic brain-injured patients were recruited for this study as participants. Patients were included if their age was higher than 18 years, if they spoke Italian native language, and if their general cognitive functioning was within the normal range at the Mini-Mental State Examination (MMSE) and at the Raven’s matrices [79,80]. Moreover, patients were included in the sample only if no comorbidities with other psychiatric or neurological conditions (anxiety, depression-related, or dementia) were detected. Eleven patients could not be included in the final sample because the number of missing measures for each of them was higher than 30%. In these patients, some tests were not administered mainly because of problems in understanding verbal instructions. Performances of the remaining 28 patients were compared by dividing them into three groups. The first group consisted of 16 patients with lesions in anterior cortical regions, i.e., frontal and/or polar areas of temporal lobe (Anterior group); the second group consisted of 6 patients with lesions in posterior cortical regions, i.e., parietal and/or posterior regions of temporal lobe (Posterior group); and the third group consisted of 6 patients with damage in both anterior and posterior brain regions (AnteroPost group). The etiology was traumatic for 14 of them and vascular for the other 12. Brain lesion localizations have been derived from CT scans (Table 2 and Figure 1).

All patients were assessed in the post-acute phase of disease, and, in the time window of the assessment, none of them were under pharmacological treatment that could alter their cognitive performance. Anamnestic characteristics did not significantly differ across groups. MMSE, Raven’s matrices, age, and years of education were compared through a series of paired *t*-tests. The *p*-values were not significant, being higher than 0.08. The distribution of gender was checked through chi-squared and the exact test of Fisher, whose highest value was χ^2^(1) = 3.38, *p* = 0.07.

### 2.2. Materials and Procedure

Executive functions have been assessed by means of a selection of the tests listed in Table 1.

Learning and memory abilities were assessed by means of the digit span test (backward) [14], the Updating Working Memory test [82], the Memory with Interference tests [83], and the Corsi’s Block Tapping test [20]. Attentional Matrices [22] and the Trail Making test, part A (TMT-A: [29]) were used to assess visual search and selective visual attention. Switching abilities were evaluated through the Trail Making test, part B (TMT-B [29]) and the WCST [48]. Cognitive flexibility was investigated with the Word Fluency test (both phonological and semantic fluency) [33] and with the Overlapping Figures test [84]. Inhibition was assessed by means of the Stroop Color-Word Test [36] and a computerized version of the Go/No-Go test. The Tower of London [45] and the Elithorn’s Perceptual Maze [43] tests have been included as measures of planning. In addition, the City Map [60] and Maps test [62] provided further measures about visuo-spatial planning. The last two tests were included to increase the reliability of planning, to face criticism elicited by the scientific literature [57]. The City Map and the Maps task present a traveling-salesperson problem presented through paper-and-pencil and computerized modalities, respectively. In the City Map, a series of 10 locations is presented on a sheet, and the participants must indicate on the city map the shortest path to visit each location once by respecting constraints (like shop opening hours). The Maps task presents 12 situations, in each of which the participant should press the keyboard’s arrow keys to move a cursor from a starting point to the end point, passing over each of the intermediate locations. These vary in number (from 4 to 9) and positions across the 12 situations. These planning tasks offer a series of different measures related to the planning performance, with respect to the other planning tasks.

A set of non-executive functions was also assessed as control measures, which are expected not to produce differences across groups. A measure of speed of processing was obtained from simple and choice reaction time tasks, whose order of administration was randomized. In these tasks, a sound was followed by a cross at the center of the screen. Afterwards, a target appeared, different depending on the type of task. In the simple RTs task, a green circle appeared at the center of the screen, and patients were required to respond as soon as possible to it by pressing a key. In the choice RTs task, a colored circle appeared, and patients were required to press the left-arrow key if the circle was red, or press the right-arrow key if it was green. Short- and long-term memory was assessed by means of the Story Recall test [22]. In the Story Recall test, patients are asked to listen to a short text, and to recall as much information as they can about the text, immediately after the examiner finishes reading it (Immediate recall), and 5 min later without being forewarned (Delayed recall). The Clock Drawing test [85] and the Rey’s Complex Figure [25] were administered to obtain a measure of visuo-spatial perception.

Participants were tested individually during the clinical assessment for diagnostic purposes in the hospital facilities. The order in which the tests were administered was counterbalanced across participants to avoid biases in the data. The assessment was in one or two sessions and lasted on a range between 1.5 and 3.5 h. During the session(s), patients were offered the chance to take little breaks to rest. Most tests were administered using paper-and-pencil modality, while four tasks (simple RTs, choice RTs, go/no-go, and the Maps task) were administered using a desktop computer (equipped with a 13-inch monitor and a standard keyboard).

### 2.3. Data Analysis

All analyses were conducted using R version 4.4.2 [86]. Performance on various cognitive tests was recorded for participants across the three groups (Anterior, Posterior, and AnteroPost). For some participants, 11 values were not available because the transcription of the data was corrupted. In these instances, missing values were imputed by replacing them with the means of the corresponding group. It should be noted that, while this is a straightforward method for handling missing data, it has the limitation of potentially underestimating variance [87]. To examine performance differences among the three groups, most tests were analyzed using a series of standard linear regression models. However, for certain measures, a GLM was more appropriate to account for the specific distribution of the data. All dependent variables representing response times (Mean RT Go/No-go, Mean Simple RT, Mean Choice RT) were analyzed using a Gamma GLM with a log link. Dependent variables based on counts (City Map: Readings) were analyzed using a Poisson GLM with a log link. Binary dependent variables (Constraints, Omissions, Perseverations) were analyzed using a binomial GLM with a logit link. Finally, the variable Errands/Movements, being a proportion, was analyzed using a beta regression model. In all models, the Anterior group was used as the reference level, meaning that comparisons were made between the Anterior and the Posterior groups and between the Anterior and the AnteroPost groups. Because different types of models were applied, the interpretation of the coefficients and test statistics reported in Table 3 varies accordingly. For the classical linear regression models, the coefficient for a categorical variable reflects the difference in means between two levels. In the Gamma GLM (log link), the coefficients indicate changes in the log of the mean response time compared to the reference level; exponentiating these coefficients yields the multiplicative effect on the mean response time. In both models, the test statistics reported are t-values. For the Poisson GLM (log link), coefficients are also on a log scale; exponentiating them gives the IRR, which represents the multiplicative change in the expected count. For the binomial GLM (logit link), exponentiating the coefficients provides the odds ratio (OR), indicating how many times the odds of success change in each level compared to the reference level. In the beta regression model, the exponentiated coefficients represent the multiplicative change in the odds associated with the expected proportion. For these three models, the test statistic reported is a z-value.

## 3. Results

Results for the executive function tests are presented in Table 3, while results for the other cognitive function tests are presented in Table 4. For most executive function tests, no significant group effects were observed. However, some tests showed significant differences between groups. In the Memory with Interference test (10 s of counting), the Anterior group scored significantly higher than the AnteroPost group (*t*_25_ = 2.71, *p* = 0.01), whereas in the Memory with Interference (30 s of counting), it scored significantly higher than the Posterior group (*t*_25_ = 2.19, *p* = 0.04). In the Attentional Matrices, the Posterior and the AnteroPost group scored significantly lower than the Anterior group (*t*_25_ = −2.17, *p* = 0.04; *t*_25_ = −3.06, *p* = 0.01). In Elithorn’s Perceptual Maze Test, the Posterior group showed significantly lower scores compared to the Anterior group (*t*_25_ = −3.00, *p* = 0.01). In the Reading measure of the City Map, the Posterior group showed significant lower scores compared to the Anterior group (Z = −4.21, *p* < 0.001). In the Planning Index Variance measure of the Maps task, the AnteroPost group showed significantly higher scores compared to the Anterior group (*t*_25_ = 2.63, *p* = 0.01). For the other cognitive functions, the only test that showed a significant group effect was the Clock Drawing test, where the AnteroPost group scored significantly lower than the Anterior group (*t*_25_ = −2.15, *p* = 0.04). The results described above are based on unadjusted *p*-values. To control for the inflation of Type I error, we subsequently applied the False Discovery Rate (FDR) correction method [88], which offers a good balance between limiting false positives and preserving statistical power. Specifically, *p*-values for executive function tests (for both Anterior vs. Posterior and Anterior vs. AnteroPost comparisons) were adjusted for 25 comparisons (the total number of executive function tests), while *p*-values for the other cognitive function tests were adjusted for 6 comparisons (the total number of those tests). After operating the correction, a significant group effect remained only for Readings in the City Map test (Anterior vs. Posterior: Z = −4.21, *p* < 0.001).

**Table 3 neurosci-06-00105-t003:** For each executive function test, the table shows the estimated coefficients, standard errors, T-value (test statistics), and *p*-values. These values are provided for both the Anterior vs. Posterior comparison (left) and the Anterior vs. AnteroPost comparison (right). An asterisk indicates *p*-values lower than alpha = 0.05.

Executive Function	Test	Anterior vs. Posterior					Anterior vs. AnteroPost				
		Coefficient	Std. Error	Test Statistics	*p*-Value	*p*-Value Corrected	Coefficient	Std. Error	Test Statistics	*p*-Value	*p*-Value Corrected
Working Memory	Digit span backward	−0.46	0.55	−0.84	0.41	0.70	0.38	0.55	0.69	0.50	0.75
	Updating working memory	−1.67	2.02	−0.82	0.42	0.70	2.17	2.02	1.07	0.29	0.73
	Memory with interference 10 s	−1.46	0.85	−1.73	0.10	0.43	−2.29	0.85	−2.71	0.01 *	0.10
	Memory with interference 30 s	−2.23	1.02	−2.19	0.04 *	0.30	−0.90	1.02	−0.88	0.39	0.75
Attention	Attentional Matrices	−10.96	5.06	−2.17	0.04 *	0.30	−15.46	5.06	−3.06	0.01 *	0.10
	TMT-A (sec.)	9.23	15.18	0.61	0.55	0.75	8.73	15.18	0.58	0.57	0.81
Inhibition	Mean RT Go/No-go (msec.)	−0.83	1.98	−0.42	0.68	0.82	0.09	1.98	0.04	0.97	1.00
Switching and Flexibility	TMT-B (sec.)	124.50	75.89	1.64	0.11	0.43	65.17	75.89	0.86	0.40	0.75
	Phonemic Fluency	−3.05	1.92	−1.59	0.13	0.43	−3.24	1.92	−1.69	0.10	0.45
	Semantic Fluency	−1.23	4.38	−0.28	0.78	0.87	−3.16	4.38	−0.72	0.48	0.75
	Overlapping Figures	0.15	0.12	1.28	0.21	0.57	0.09	0.12	0.74	0.47	0.75
Planning	Tower of London	−1.42	2.83	−0.50	0.62	0.78	−1.02	2.83	−0.36	0.72	0.90
	Elithorn’s Perceptual Maze Test	−6.53	2.18	−3.00	0.01 *	0.15	−1.95	2.18	−0.89	0.28	0.73
	City Map test:										
	Errands/movements	−0.69	0.64	−1.08	0.28	0.60	0.25	0.58	0.43	0.67	0.90
	Total time	−2.53	2.18	−1.16	0.26	0.60	3.50	2.18	1.60	0.12	0.45
	Readings	−0.82	0.19	−4.21	<0.001 *	<0.001 *	0.05	0.14	0.39	0.70	0.90
	Constraints	0.77	1.08	0.72	0.47	0.74	−0.14	1.27	−0.11	0.91	1.00
	Omissions	0.10	1.02	0.09	0.93	0.95	0.79	0.98	0.81	0.42	0.75
	Perseverations	0.69	1.00	0.69	0.49	0.74	0.00	0.96	0.00	1.00	1.00
	Maps test:										
	Optimization index	0.90	0.65	1.38	0.18	0.54	0.98	0.65	1.51	0.15	0.50
	Initial Planning Time (msec.)	1931	2926	0.66	0.52	0.74	6060	2926	2.07	0.05	0.38
	Execution time (msec.)	13,480	8654	1.56	0.13	0.43	7727	8654	0.89	0.38	0.75
	Planning Index	0.98	0.97	1.01	0.32	0.60	1.60	0.97	1.64	0.11	0.45
	Planning Index Variance	1.11	3.07	0.36	0.72	0.83	8.06	3.07	2.63	0.01 *	0.10
	Heuristic dispersion	2.69	5.05	0.53	0.60	0.78	6.41	6.09	1.05	0.30	0.82

**Table 4 neurosci-06-00105-t004:** For each test, the table shows the estimated coefficients, standard errors, T-value (test statistics), *p*-values, and FDR-corrected *p*-values. These values are provided for both the Anterior vs. Posterior comparison (left) and the Anterior vs. AnteroPost comparison (right). An asterisk indicates *p*-values lower than alpha = 0.05.

Other Cognitive Functions	Test	Anterior vs. Posterior				
		Coefficient	Std. Error	T-Value	*p*-Value	*p*-Value Corrected
Speed of processing	Mean Simple RT (msec.)	0.16	0.14	1.16	0.26	0.39
	Mean Choice RT (msec.)	0.20	0.10	2.06	0.05	0.30
Short-term memory	Story recall (Immediate)	−3.97	2.29	−1.73	0.10	0.30
	Story recall (Delayed)	−3.67	2.60	−1.41	0.17	0.34
Visuo-spatial abilities	Clock Drawing	−0.91	1.15	−0.79	0.44	0.53
	Rey’s Complex Fig. Copy	−0.38	0.90	−0.42	0.68	0.68
		**Anterior vs. AnteroPost**				
Speed of processing	Mean Simple RT (msec.)	0.11	0.14	0.82	0.42	0.50
	Mean Choice RT (msec.)	0.09	0.10	0.98	0.34	0.50
Short-term memory	Story recall (Immediate)	0.87	2.29	0.38	0.71	0.71
	Story recall (Delayed)	3.33	2.60	1.28	0.21	0.42
Visuo-spatial abilities	Clock Drawing	−2.48	1.15	−2.15	0.04 *	0.24
	Rey’s Complex Fig. Copy	−1.61	0.90	−1.79	0.09	0.27

Three figures illustrating the effect of group on each dependent variable (i.e., each test score) are provided in Appendix A (Figure A1, Figure A2 and Figure A3, for EF test, EF planning tests, and non-EF tests, respectively). A raincloud plot is used for all variables except for the binary dependent variables. For those variables, the plots show the probability of success with the corresponding standard errors for each group.

## 4. Discussion

The present study was aimed at providing further data on the disputed triadic relationship among EFs, EF tests, and frontal associative brain regions. To this end, three groups of patients with brain lesions located in either anterior, posterior, or antero-posterior regions were examined. Several tests measuring executive functions were used to clinically assess whether this relationship could be confirmed. Our results indicate that only a few tests showed significant differences in scores between the three lesion groups (namely the Memory with Interference test, the Attentional Matrices, and some measures of planning). However, these results were significant only when the tests were considered independently. In fact, when corrected for multiple comparisons, just one significant difference survived: the frequency with which the list of locations was read in the City Map task. This measure is associated with the organization of the plan, but it is a secondary index with respect to other ones. Therefore, it is likely that the significance of this variable may be due to a simple casual fluctuation, while the result is that no other measure achieved a significant level.

The results of our study suggest that the tests most used in clinical research to discriminate between brain lesions are not specific enough. Our findings are in line with those of previous literature, such as [68], which stated: “Psychometric tests for EF also show sensitivity, but not specificity for frontal lobe damage” (p. 3). One problem related to tests can be due to the administration modality, as suggested by [89]: they observed that paper-and-pencil tests may lack sensitivity to estimate EFs. The use of computerized testing could overcome this limitation by capturing more variable or subtle deficits than traditional paper-and-pencil tests [90]. The measurement of EFs was initially debated due to concerns about the external validity [91], but recent studies are revealing a deeper bias. In accordance with this, our graphs show that not only the average scores of the measures were not different between the anterior and posterior groups, but also that the across-patients distribution and variability of these measures were largely overlapping. This evidence further supports the criticisms about the association between EFs and the tests used to measure them.

By addressing a more central concern, this study provides evidence that the relationship between EFs and brain regions should be reconsidered, and that the notion of EFs being centered in the frontal associative regions is weakened. These criticisms are not new, given that past studies have shown that the frontal areas are not the sole neural substrate of EFs [6,53]. Research involving functions such as working memory [54], inhibition [92], and cognitive flexibility [93] has demonstrated that these processes rely on broader brain networks beyond the prefrontal cortex alone. The concept of EFs was initially defined by Lezak [94] as “those mental capacities necessary for formulating goals, planning how to achieve them, and carrying out the plans effectively” (p. 281). Although in this definition there was no mention of brain areas, in the same paper, the author highlighted the relationship between EFs and PFC areas. Interestingly, she also stated that “damage to other areas of the brain can also interfere with executive function” (p. 284). Therefore, the strict association between EFs and frontal areas could be regarded as a myth or, at best, a trivialization of the problem.

The current study also has some limitations that should be acknowledged. First, the number of patients was relatively small, and this could have led to hiding significant effects. Although modest, the inclusion of 28 patients is consistent with the average sample size in clinical neuropsychology research. The etiology was not homogeneous across patients, and this may have added variability to the data. Notwithstanding, a higher number of patients would be desirable in future studies to test whether these results could be generalized. To this aim, the transcranial magnetic stimulation (TMS) and transcranial direct current stimulation (tDCS) are useful techniques to simulate local impairment and could help to improve our understanding of the relationship between localized brain areas and their associated EF. However, they rarely reflect a real brain injury, which is likely to entail broader and irregular portions of the brain. A brilliant opportunity is offered by the recent trend in creating multi-lab studies, which could share methodology and collect higher numbers of participants. The only problem with respect to its application to patients consists of the comparison of the results, which will be obtained by tests administered using different languages. Intercultural studies should be conducted to validate measures across countries. Another limitation, which may affect the interpretation of our results, consists of the chosen tests. Although they were selected from those more commonly used in the relevant literature, they are mainly used for clinical studies, rather than to study the cognitive processes. Thus, one aspect of the triadic relationship between EFs, brain regions, and tests cannot be convincingly questioned. Many scholars would argue that if EFs were tested with proper measures, results could have been different [95]. However, the issues related to content validity are very challenging, even more so when the theoretical concept is still under debate, like that of EFs.

Based on these remarks, some recommendations for the clinical community could be devised. The tasks used to assess EFs require either improvement or deeper analysis. While clinical assessment requires optimization between time and precision, theoretical knowledge does not have such constraints. Therefore, clinical neuropsychologists are encouraged to adopt more refined methodological approaches to define what is measured by the existing EF tests. This endeavor seems more feasible (and cheap) than ideating further paradigms more suitable than the existing ones, although both of them are subjected to the admonition by [96]: “various types of behaviors/deficits exhibited through the performance measures of different executive tasks are just that—the expression of different executive behaviors—and not necessarily a reflection of multiple separate executive ‘functions’.” (p. 466). Secondly, EFs and frontal areas should not be meant as synonyms. There is a certain level of overlapping, but keeping separate the concept of EFs and brain regions would avoid equivocation in those cases in which the two aspects do not co-occur. Obviously, scholars working in the field of neuropsychology are not concerned by this caveat: the level of precision used to define experimental methodologies and to identify brain regions responsible for certain functions is so high that there is no need to use these two umbrella terms. The recommendation is aimed at students, trainees, laypersons, journalists, at anyone who can come into contact with the concept, being however not prepared enough to handle it.

## 5. Conclusions

In conclusion, the present paper provides data against the supposed relationship between executive functions, their brain localization, and their measurement. While the tests used to study EFs are not specific enough, research is invited to validate tests with higher statistical power. Moreover, given that the relationship between EFs and brain regions is currently weak, we believe that a more tailored concept of EFs would help both researchers and practitioners. We do not imply that the concept of EFs to be dismissed, because of its importance for the study of cognitive processes and for the clarity required for the dissemination of results. Rather, it seems to be beneficial to move toward two improvements: (a) development of new robust tests, which can satisfy statistical standards, and (b) network-based theorization, not based on a static one-to-one relationship between EFs and brain regions. While this paradigm shift takes place, the results of the current study emphasize practitioners to be cautious when establishing strict links between the three aspects of the EFs concept: its definition, its measurement, and its localization in the brain.

## Figures and Tables

**Figure 1 neurosci-06-00105-f001:**
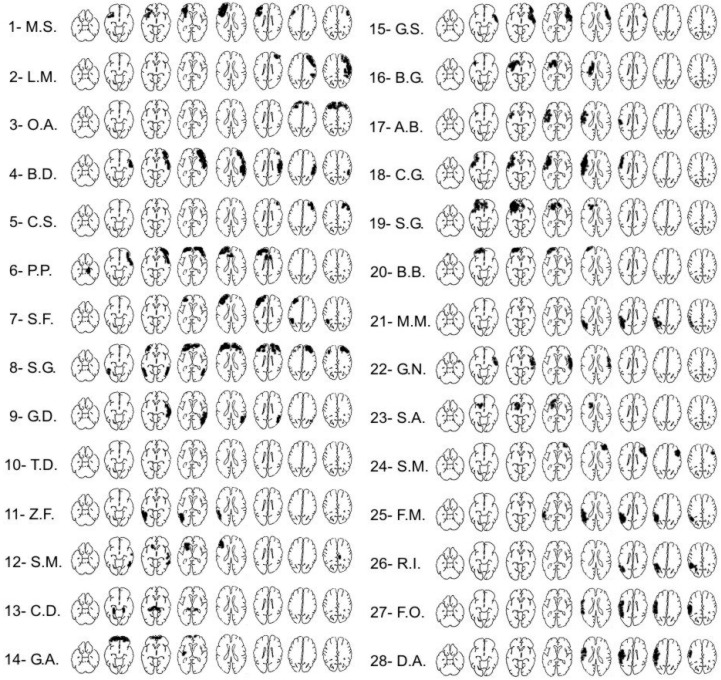
Reconstruction of the lesions’ locations for the 28 patients, based on CT scans. For each patient, the horizontal slices, from left to right, go from most inferior to most superior. The right side of each slice corresponds to the right side of the brain.

**Table 1 neurosci-06-00105-t001:** The table presents several EFs (first column), with a list of the most common tests used to test them (second column). The paper presenting the task in scientific literature is provided in the third column.

Processes	Measures	Source
Learning and Memory	- Digit Span	Wechsler (1945) [14]
	- Self-Ordered Pointing	Petrides & Milner (1982) [15]
	- Sequential Matching Memory test	Gorestein et al. (1989) [16]
	- Temporal Order Memory	McAndrews & Milner (1991) [17]
	- Source Memory	Schacter (1987) [18]
	- California Verbal Learning Test	Delis et al. (1987) [19]
	- Corsi Block Tapping	Corsi (1971) [20]
	- N-back paradigm	Gevins and Cutillo (1993) [21]
	- Story recall	Spinnler & Tognoni (1987) [22]
	- Updating working memory	Morris & Jones (1990) [23]
	- Memory with interference	Belleville, Peretz & Malefant (1996) [24]
	- Rey-Osterrieth’s complex figure	Osterrieth (1944) [25]
Attention	- Test of Everyday Attention for Children	Manly et al. (2001) [26]
	- Continuous Performance Test	Conners (1992) [27]
	- Attentional Matrices	Spinnler & Tognoni (1987) [22]
	- Cancellation test	Toulouse & Pieròn (1904) [28]
	- Trail Making test, part A	Reitan (1958) [29]
Switching and Flexibility	- Wisconsin Card Sorting test	Milner (1963) [30]
	- Trail Making test, part B	Reitan (1958) [29]
	- Hayling Sentence Completion	Burgess & Shallice (1996) [31]
	- Brixton Spatial Anticipation test	Burgess & Shallice (1997) [32]
	- Word Fluency	Borkowsky et al. (1967) [33]
	- Design Fluency	Jones-Gotman & Milner (1977) [34]
	- Overlapping Figures	Poppelreuter (1917) [35]
Inhibition	- Stroop test	Stroop (1935) [36]
	- Go/No-go	Luria & Homskaya (1964) [37]
	- Stop signal	Logan (1994) [38]
	- Negative priming	Tipper (1985) [39]
	- Anti-saccade task	Guitton et al. (1985) [40]
	- Conflicting Motor Response	Wright et al. (2003) [41]
Planning	- Porteus Mazes	Porteus (1959) [42]
	- Elithorn’s Perceptual Maze	Elithorn (1955) [43]
	- Tower of Hanoi	Borys et al. (1982) [44]
	- Tower of London	Shallice (1982) [45]

**Table 2 neurosci-06-00105-t002:** For each patient, an “X” is present if the brain region shows impairment. Correspondence between Damasio and Damasio codes [81] and Broadmann areas is: F01, anterior cingular gyrus (23); F02, posterior cingular gyrus (22,30); F03, supplementary motor area (6); F04, prefrontal area (8–10); F05, rolandic region (1–4); F06, frontal operculum (43,44); F07, prefrontal region (8,9,45); F08, premotor region (6) and rolandic region (1–4); F09, paraventricular; F10, supraventricular area; F11, anterior orbital; F12, posterior orbital, T, temporal regions; P, parietal regions.

	Left Hemisphere														Right Hemisphere													
Patients	F01	F02	F03	F04	F05	F06	F07	F08	F09	F10	F11	F12	T	P	F0	F02	F03	F04	F05	F06	F07	F08	F09	F10	F11	F12	T	P
1- M.S.	X			X		X	X		X		X	X	X				X											
2- L.M.																		X		X	X	X						X
3- O.A.				X			X											X			X							
4- B.D.																		X		X	X	X	X				X	X
5- C.S.																		X		X	X							
6- P.P.	X			X			X		X	X					X			X		X	X		X	X		X		
7- S.F.					X	X	X						X	X														
8- S.G.				X			X						X	X				X			X						X	X
9- G.D.																				X							X	X
10- T.D.		X														X												
11- Z.F.													X	X									X					
12- S.M.						X			X										X								X	
13- C.D.		X							X							X							X					
14- G.A.				X		X	X				X							X			X				X			
15- G.S.																			X	X	X							
16- B.G.	X					X					X																	
17- A.B.						X		X																				
18- C.G.						X	X	X																				
19- S.G.	X				X	X	X		X						X								X					
20- B.B.				X			X				X																	
21- M.M.					X								X	X														
22- G.N.																				X		X					X	
23- S.A.	X								X			X																
24- S.M.																		X		X	X	X						
25- F.M.													X	X														
26- R.I.														X														
27- F.O.						X		X						X														
28- D.A.						X		X						X														

## Data Availability

The original data presented in the study are openly available in the OSF repository of the current project (2cehu), at: https://osf.io/2cehu/overview?view_only=d5f05a71989742b197c27f1fea074a5f (accessed on 11 October 2025).

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
