# Peer review of "Frontal Regions and Executive Function Testing: A Doubted Association Shown by Brain-Injured Patients"

_neurosci, 2025, doi:10.3390/neurosci6040105_

Round 1

Reviewer 1 Report

Comments and Suggestions for Authors

The study is interesting and a welcome addition to the literature. Below are some minor points to increase the clarity of the paper.

  • "coordinate simpler cognitive abilities". Give some examples of such simpler cognitive abilities.
  • table: most of the sources are rather old. Are there no more recent sources? if not, this is woth noting.
  • add details what 'Stroop performance' is.
  • "of a common underlying mechanism". What is this mechanism?
  • what do you mean by "ultimate results"?
  • It would be interesting if the studies using VR also provide evidence against localizing executive function in the designated frontal associative brain areas. This can be added to section 1
  • “cannot be reduced to a single lesion or mechanism”. The way it’s phrased is a bit misleading. There might be multiple lessions or mechanisms within the frontal associative brain areas.
  • “Although the competing views remain controversial on many points”. Add references to criticisms.
  • 2 add whether additional checks for attention were included.
  • It would help to state the tested hypothesis more clearly.
  •  

Reviewer 2 Report

Comments and Suggestions for Authors

The authors have prepared a well-written article. Here are some points to improve the article: The introduction is too long. In different parts, it is written as a literature review rather than an introduction. The knowledge gap and the research goal are unclear. Some ideas are repeated several times.

The methods section needs more detailed inclusion/exclusion criteria. While age, language, and cognitive scores are mentioned, crucial comorbidities like depression, anxiety, prior cognitive issues, and psychotropic medication use are omitted. The extensive test battery (1.5-3.5 hours) raises concerns about participant fatigue, despite reported rest periods. The order of test administration, a critical factor in neuropsychological studies due to learning and fatigue effects, is not specified. Furthermore, the validity and reliability of the instruments should be included. In the results section, reporting individual t and Z values in the text has caused weight and reduced readability. Usually, this information should be presented in tables and only the main findings and general patterns should be reported in the text.

The discussion section unnecessarily repeats the results (Memory with interference, Attentional Matrices, City Map). Rather than repeating statistics, the discussion should focus on the overall patterns and the interpretation of the results' meaning, with explanations provided by the authors' analysis and interpretation.

The conclusion is too vague and requires substantial revision to articulate a clear and definitive position.
